# AUM302, a novel triple kinase PIM/PI3K/mTOR inhibitor, is a potent *in vitro* pancreatic cancer growth inhibitor

Komala Ingle[1], Joseph F. LaComb[1], Lee M. Graves[2], Antonio T. Baines[2,3], Agnieszka B. Bialkowska[1]*

1 Department of Medicine, Renaissance School of Medicine at Stony Brook University, Stony Brook, New York, United States of America, 2 Department of Pharmacology, School of Medicine, the University of North Carolina at Chapel Hill, Chapel Hill, North Carolina, United States of America, 3 Department of Biological & Biomedical Sciences, College of Health & Sciences, North Carolina Central University, Durham, North Carolina, United States of America

* Agnieszka.Bialkowska@stonybrookmedicine.edu

**Data Availability Statement:** All relevant data are within the manuscript and its Supporting information files.

## Abstract

Pancreatic cancer is one of the leading causes of cancer deaths, with pancreatic ductal adenocarcinoma (PDAC) being the most common subtype. Advanced stage diagnosis of PDAC is common, causing limited treatment opportunities. Gemcitabine is a frequently used chemotherapeutic agent which can be used as a monotherapy or in combination. However, tumors often develop resistance to gemcitabine. Previous studies show that the proto-oncogene PIM kinases (PIM1 and PIM3) are upregulated in PDAC compared to matched normal tissue and are related to chemoresistance and PDAC cell growth. The PIM kinases are also involved in the PI3K/AKT/mTOR pathway to promote cell survival. In this study, we evaluate the effect of the novel multikinase PIM/PI3K/mTOR inhibitor, AUM302, and commercially available PIM inhibitor, TP-3654. Using five human PDAC cell lines, we found AUM302 to be a potent inhibitor of cell proliferation, cell viability, cell cycle progression, and phosphoprotein expression, while TP-3654 was less effective. Significantly, AUM302 had a strong impact on the viability of gemcitabine-resistant PDAC cells. Taken together, these results demonstrate that AUM302 exhibits antitumor activity in human PDAC cells and thus has the potential to be an effective drug for PDAC therapy.

## Introduction

Pancreatic cancer is the seventh leading cause of cancer deaths worldwide and the third leading cause of cancer deaths in the United States and its predicated to become the second leading cause of death by 2030 [1, 2]. Pancreatic cancer has a poor prognosis, with a 5-year survival rate of just 12% [1, 3]. This low survival rate is caused by several factors, of which perhaps the most important is the prevalence of late-stage diagnoses, with 80% of patients having locally advanced or metastatic pancreatic cancer at the time of diagnosis [4]. Current treatment options include surgical resection, chemotherapy, and radiotherapy [4]. However, only 10%-20% of patients are eligible for curative resection [5] and tumors often show resistance to

**Funding:** The work was supported by a grant from the National Institutes of Health awarded to ABB (DK124342). The funders had no role in study design, data collection and analysis, decision to publish, or preparation of the manuscript.

**Competing interests:** The authors have declared that no competing interests exist.

chemotherapy and radiotherapy [6, 7]. Gemcitabine is a widely used chemotherapeutic agent against locally advanced and metastatic pancreatic cancer [8–10]. Although pancreatic cancer is most receptive to gemcitabine than other anticancer agents, many patients develop resistance within weeks of starting the treatment [11]. Multiple studies showed that reactivation and/or deregulation of SHH, PI3K/AKT, MEK, WNT, and NOTCH signaling pathways impact cell cycle and apoptosis and, in combination with disruption of gemcitabine metabolism leads to development of gemcitabine resistance [12–18]. Recent studies showed that remodeling of gemcitabine metabolism pathway and targeting apoptotic machinery provide promising results [8, 19–21].

Pancreatic ductal adenocarcinoma (PDAC), the most common type of pancreatic cancer, is genetically heterogeneous driven by mutations in oncogenes and tumor suppressor. K-Ras mutations are major drivers of PDAC development and progression and are identified in 90% of PDAC cases [22–25]. Mutations in tumor-suppressor genes such as *CDKN2A*, *TP53*, or *SMAD4*, and in oncogenes *ERBB2* and *EGFR*, and in signaling pathways genes and genes regulating metabolism accelerate the formation and progression of pancreatic lesions [23, 24, 26, 27]. Dysregulation of multiple signaling pathways allows tumor cells to resist cell death, increase angiogenesis, invasion and metastasis, modify metabolism to nutrient- and oxygen-deficient environment, and remodel tumor-promoting immune response [25, 28–30].

Proviral integration site for moloney murine leukemia virus kinases (PIMs) are serine/threonine kinases that promote cell survival by regulating the cell cycle, cell proliferation, apoptosis, and transcription [31–33]. The PIM family consists of three members, PIM1, PIM2, and PIM3, from which PIM1 and PIM3 have been shown to be upregulated in solid cancers, while PIM2 mostly in hematological cancers [33–36]. PIM1 is upregulated in primary pancreatic tumor tissue compared to matched normal tissue in PDAC patients due to hypoxic environment and has been identified as prognostic marker [37, 38]. A study by Li and colleagues found PIM3 abundantly expressed in pancreatic cancer tissue but not in normal pancreatic tissue [32]. The overexpression of the PIM kinase family is related to chemotherapy and radiotherapy resistance with PIM3 expression specifically acting as a prognostic indicator related to poor patient survival [39]. PIM1 increases the stability of c-Myc through phosphorylation and together they promote cell cycle progression [40]. Multiple studies demonstrated that selective PIM inhibitors reduce phosphorylation levels of ribosomal protein S6 and thus, modulate the translation potential of numerous cancer cell lines [41–45]. PIM1 and PIM3 can phosphorylate pro-apoptotic BAD at Ser-112 to deactivate it and thereby promote cancer cell survival and progression [40]. Importantly, studies show that inhibition of PIM1 or PMI3 in PDAC cells reduces growth, invasion, and chemosensitizes the cells to gemcitabine treatment, respectively [37, 46]. Furthermore, PIM kinases interact with the PI3K/AKT/mTOR pathway to drive cancer cell proliferation and survival [31, 47]. The regulation of mTOR signaling by PIM can also affect mTOR outputs such as S6 kinase affecting cell growth and metabolism [48]. Consequently, PIM kinases are appropriate targets for cancer therapy through the use of PIM kinase inhibitors.

TP-3654 is a second generation small-molecule PIM kinase inhibitor that has been studied *in vitro* in several cancers, including pancreatic cancer, and is currently being used in a Phase I first-in-human study in patients with advanced solid tumors [49, 50]. The compound AUM302 is a novel triple PIM/PI3K/mTOR inhibitor that has been shown to induce apoptosis and decrease cell viability in prostate cancer [51]. Co-targeting of PIM and PI3K/AKT/mTOR pathways may be a useful approach as these kinases share several downstream targets such as p21, p27, and BAD [47]. Importantly, PI3K pathway has been implicated as one of the mechanisms of gemcitabine resistance in PDAC and targeting its activity provided potential path to sensitize the cells to gemcitabine [13, 52–56]. The aim of this study is to determine the efficacy

of AUM302 in comparison with TP-3654 and gemcitabine in inhibiting pancreatic cancer cell lines growth. Here, we show that AUM302, a novel triple kinase PIM/PI3K/mTOR inhibitor, decreases proliferation of pancreatic cancer cell lines *in vitro*. Moreover, we showed that AUM302 sensitized pancreatic cancer cells' response to gemcitabine treatment.

## Materials and methods

### Cell lines and compounds

PDAC cell lines BxPC-3 (CRL-1687), Capan-2 (HTB-80), MIA PaCa-2 (CRL-1420), PANC-1 (CRL-1469), and Hs766T (HTB-134) were purchased from ATCC (Manassas, VA) in 2020 and 2021. BxPC-3 cells were maintained in RPMI-1640 medium and Capan-2 cells in McCoy's medium. MIA PaCa-2, Hs766T and PANC-1 cells were maintained in DMEM medium. All media were supplemented with 10% FBS and 1% penicillin/streptomycin. MIA PaCa-2-Gemcitabine (MIA PaCa-2 GemR) resistant cells were a gift from the laboratory of Dr. Lee M. Graves (Department of Pharmacology, School of Medicine, the University of North Carolina at Chapel Hill, Chapel Hill, NC, USA). (MIA PaCa-2 GemR) cells were grown in DMEM medium supplemented with 10% FBS and 1% penicillin/streptomycin and 50 nM gemcitabine. All cells were maintained at 37˚C and 5% $CO_2$. All experiments were performed on cells with passage range between 5 and 29. TP-3654, GDC-0941, BEZ235, and gemcitabine were purchased from Selleck Chemicals (Houston, TX) and AUM302 was provided by AUM Biosciences. TP-3654, AUM302, GDC-0941, BEZ235 and gemcitabine were suspended in DMSO.

### Cell viability assay

BxPC-3, Capan-2, MIA PaCa-2, PANC-1, and Hs766T pancreatic cancer cells were seeded at 1 x $10^3$ cells per well in 100 µl of appropriate media in 96-well plate format. Twenty-four hours post seeding, cells were treated with DMSO (vehicle) and variable concentrations of TP-3654, AUM302, GDC-0941, BEZ235 and gemcitabine for 72 hours. MIA PaCa-2 GemS and MIA PaCa-2 GemR cell lines were seeded at 1 x $10^3$ cells per well in 100 µl of appropriate media in 96-well plate format, treated with 10 nM, 100 nM, or 1 µM of TP-3654 or AUM302 for 72 hours. Cell viability was analyzed using the Cell Titer-Glo luciferase assay system (Promega; Madison, WI), according to the manufacturer's protocol, and a SpectraMax M3 plate reader (Molecular Devices; San Jose, CA). The $IC_{50}$ values were calculated using GraphPad Prism for Windows version 10.0.2 (GraphPad Software) [57–60].

### Cell proliferation assay

BxPC-3, Capan-2, MIA PaCa-2, PANC-1, and Hs766T pancreatic cancer cells were seeded at 7.5 x $10^4$ cells per well in 2 ml of appropriate media in 6-well plate format. Twenty-four hours after seeding, cells were treated with DMSO (vehicle) or TP-3654 (10 nM and 100 nM) or AUM302 (10 nM and 100 nM). MIA PaCa-2 GemR cell line was seeded as mentioned above, and then treated with 10 nM, 100 nM, or 1µM of TP-3654 or AUM302. Cell count was determined using the Z-Series Coulter Counter (Beckman Coulter; Indianapolis, IN) after 24, 48, and 72 hours of treatment. Each experiment was performed in triplicate. The measurement of the control (cells with medium and DMSO) was defined as 100% and the results from other measurements were calculated accordingly [57–60].

### Cell cycle assay

BxPC-3, Capan-2, MIA PaCa-2, PANC-1, and Hs766T pancreatic cancer cells were seeded at 7.5 x $10^4$ cells per well in 2 ml of appropriate media in 6-well plate format. Twenty-four hours

after seeding, cells were treated with DMSO (vehicle) or TP-3654 (100 nM) or AUM302 (100 nM). Cells were stained with propidium iodide and analyzed by FACS analysis using Cytoflex PC after 24, 48, and 72 hours of treatment. Each experiment was performed in triplicate. The measurement of the control (cells with medium and DMSO) was defined as 100% and the results from other measurements were calculated accordingly [59, 60].

### Western blot analysis

BxPC-3, Capan-2, MIA PaCa-2, PANC-1, Hs766T, MIA PaCa-2 GemR pancreatic cancer cells were seeded at $7.5 \times 10^4$ cells per well in 2 ml of appropriate media in 6-well plate format. Twenty-four hours post seeding, the first five cell lines were treated with DMSO (vehicle) or TP-3654 (10 and 100 nM) or AUM302 (10 and 100 nM) for 72 hours. MIA PaCa-2 GemR cells were treated with 10 nM, 100 nM, and 1 μM of TP-3654 or AUM302 for 72 hours. Cells were lysed in Laemmli buffer and total protein extracts were subjected to electrophoresis in 4–20% or 10% tris-glycine gels. The proteins were then transferred to a nitrocellulose membrane, blocked in 5% non-fat milk in 1 x TTBS buffer, and developed with appropriate antibodies. Protein bands were detected using an enhanced chemiluminescence detection kit using Azure c400 (Azure Biosystems). Densitometry analysis of western blots was performed using FIJI software [61].

### Statistical analysis

The analysis was performed using appropriate statistical test with a value of $p < 0.05$ considered significant. This analysis was performed using GraphPad Prism for Windows version 10.0.2 (GraphPad Software).

## Results

### AUM302 is a potent inhibitor of pancreatic cancer cell lines growth *in vitro*

To assess the efficacy of gemcitabine, TP-3654, and AUM302 on the viability of pancreatic cancer cell lines, we performed cell proliferation and growth assays using BxPC-3, Capan-2, MIA PaCa-2, PANC-1, and Hs766T pancreatic cancer cells treated with variable concentrations of these compounds for 72 hours. Furthermore, we treated these cell lines with two PI3K/mTOR inhibitors, GDC-0941 and BEZ235, to establish whether AUM302 or TP-3654, triple PIM/PI3K/mTOR inhibitors have higher efficacy of growth inhibition than dual-inhibitors [62–64]. We determined $IC_{50}$ values using the Cell Titer-Glo assay (Fig 1 and Table 1). Our results showed that all five compounds inhibit the viability of tested pancreatic cancer cell lines. However, as shown in Table 1, the AUM302 compound has more favorable $IC_{50}$ values as compared to gemcitabine, GDC-0941, and BEZ235 in BxPC-3, Capan-2, PANC-1, and Hs766T pancreatic cancer cell lines with AUM302 compound being even more effective than TP-3654. Only in MIA PaCa-2 cell line (Fig 1C, Table 1) are $IC_{50}$ values for BEZ235 and GDC-0941 lower than AUM302. Further studies compared the effectiveness of two triple PIM/PI3K/mTOR inhibitors: TP-3654 and AUM302.

To examine the impact of TP-3654 and AUM302 on the growth of pancreatic cancer cell lines, we performed cell proliferation assay using previously tested pancreatic cancer cell lines. As shown in Fig 2, the two compounds, each tested at 10 nM and 100 nM, significantly inhibited proliferation and growth of pancreatic cancer cells over the course of three-day treatment in comparison to DMSO (vehicle)-treated cells. Additionally, analysis showed that AUM302 demonstrated a robust inhibitory effect, not only in comparison with vehicle but also in comparison to TP-3654 treatment in all tested pancreatic cancer cell lines (Fig 2).

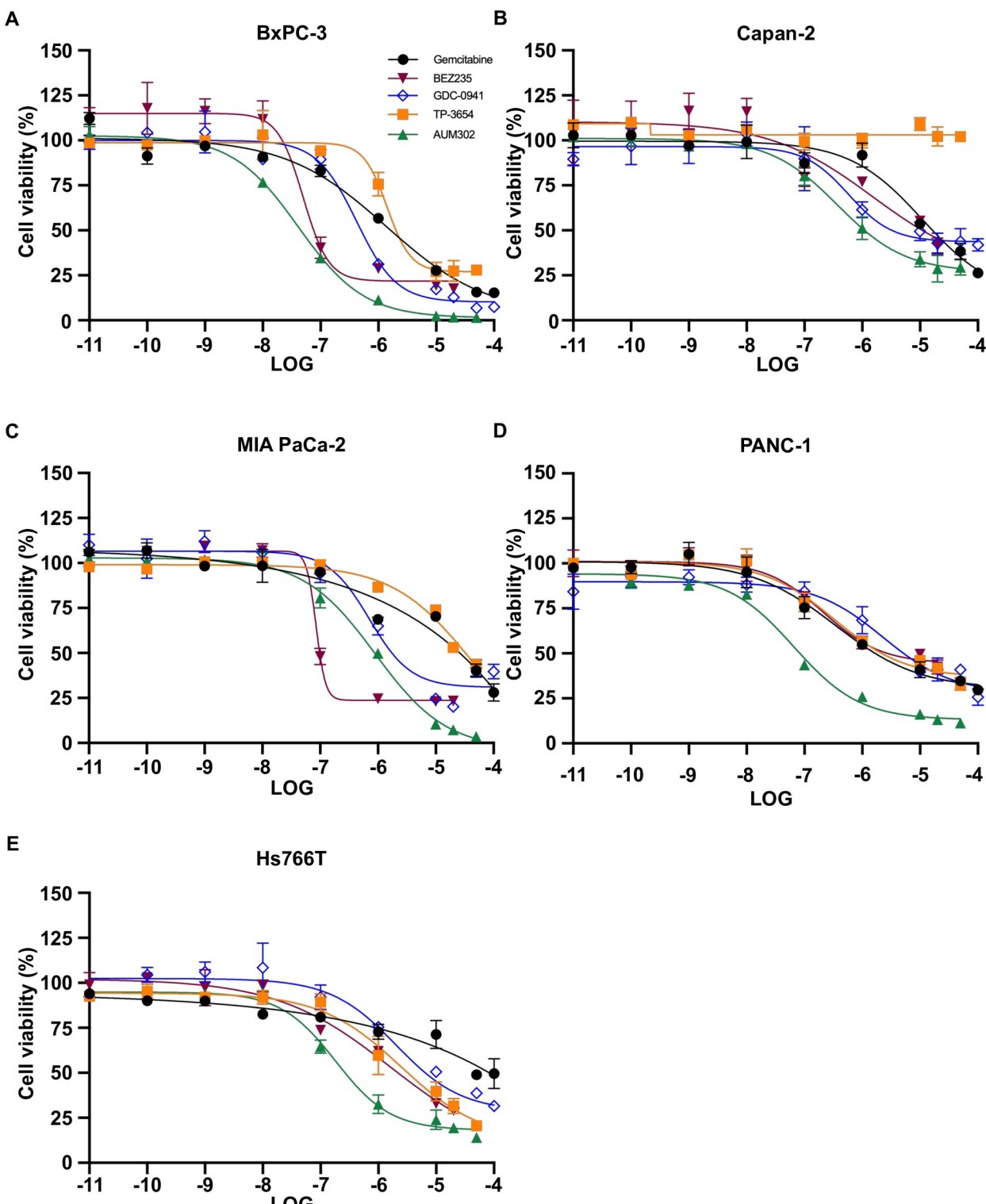

**Fig 1. Gemcitabine, BEZ235, GDC-0941, TP-3654, and AUM302 inhibit viability of multiple pancreatic cancer cell lines.** Pancreatic cancer cell lines BxPC-3 (A), Capan-2 (B), MIA PaCa-2 (C), PANC-1 (D), and Hs766T (E) were treated with variable concentrations of gemcitabine or BEZ235 or GDC-0941 or TP-3654 or AUM302 twenty-four hours after seeding. Cells were treated with test compounds for 72 hours and cell viability was measured using Cell Titer-Glo. Each experiment was performed in triplicate and the results are shown as mean ±SD (N = 3).

**Table 1. Experimental analysis of IC$_{50}$ values of gemcitabine, BEZ235, GDC-0941, TP-3654, and AUM302 tested in BxPC-3, Capan-2, MIA PaCa-2, PANC-1, and Hs766T pancreatic cancer cell lines.**

| Cell line | Gemcitabine | BEZ235 | GDC-0941 | TP-3654 | AUM302 |
|---|---|---|---|---|---|
| BxPC-3 | 1375 ± 1.59 | 49.96 ± 1.29 | 418.8 ± 1.18 | 1418 ± 1.38 | 41.12 ± 1.07 |
| Capan-2 | 11080 ± 1.99 | 1517 ± 12.42 | 576 ± 1.44 | Unstable | 376 ± 1.25 |
| MIA PaCa-2 | 5.40E+09* | 84.84* | 713.4 ± 1.29 | 40390 ± 3.46 | 891.2 ± 1.16 |
| PANC-1 | 341.1 ± 1.38 | 214.6 ± 1.32 | 2308 ± 1.60 | 333.4 ± 1.38 | 65.6 ± 1.17 |
| Hs766T | 1.57E+12* | 1653 ± 3.38 | 2021 ± 1.52 | 2494 ± 1.71 | 182.2 ± 1.16 |

The values are expressed in nanomolar concentrations as mean with ± standard error.

*—standard error of mean (SEM) LogIC50 value > 12.

## AUM302 alters cell cycle progression of pancreatic cancer cell lines

PIM kinases and PI3K/mTOR signaling pathways have been shown to play an important role in the regulation of cell cycle progression in multiple cancers, including pancreatic [18, 65–71]. Thus, we evaluated the impact of TP-3654 and AUM302 on cell cycle progression. We treated BxPC-3, Capan-2, MIA PaCa-2, PANC-1, and Hs766T pancreatic cancer cell lines over three days with vehicle or 100 nM of test compounds and then analyzed cell cycle using flow cytometry. Our results demonstrated that TP-3654 at tested concentration (Figs 3–5) does not significantly affect the cell cycle progression of tested pancreatic cancer cell lines. In contrast, AUM302 was able to increase the cell number in G$_0$/G$_1$ phases, G$_2$/M, and decrease the number of cells within the S-phase as compared to vehicle-treated cells. Notably, in the case of AUM302 there was a significant modification in the number of cells within aforementioned phases, in comparison to not only vehicle-treated cells but also to cells treated with TP-3654 compound. Importantly, in BxPC-3 and Capan-2 pancreatic cancer cell lines, treatment with AUM302 compound increased the number of cells within subG$_1$ population, suggesting that this compound may induce apoptosis (Fig 3). These data suggest that treatment with AUM302 compound alters the cell cycle of pancreatic cancer cell lines and can additionally induce cell apoptosis.

## TP-3654 and AUM302 alter the cell signaling pathways controlled by PIM kinases and PI3K/mTOR signaling pathway

PIM kinases have been shown to positively modulate gene expression in the cell cycle and inhibit apoptosis by directly and indirectly regulating multiple targets such as c-Myc, BAD, and P21 [40, 72–75]. Furthermore, studies showed that inhibitors of PIM kinases reduce the phosphorylation status of ribosomal protein S6 in multiple cancer types [76]. In addition, myriad publications showed that the PI3K/mTOR pathway plays an essential role in the tumorigenesis of numerous cancers, including pancreatic cancer [16, 77–79]. Thus, we decided to investigate the ability of TP-3654 and AUM302 to alter the expression levels of the components of these pathways in pancreatic cancer cell lines. We treated BxPC-3, Capan-2, MIA PaCa-2, PANC-1, and Hs766T pancreatic cancer cell lines with DMSO (vehicle) and tested compounds at 10 nM or 100 nM, collecting the cells for western blot analysis at 24h. As shown in Fig 6, TP-3654 and AUM302 alter the expression of analyzed proteins. However, the AUM302 compound has the most negative effect on the phosphorylation of mTOR, AKT, and S6 kinases (Figs 6 and 7) while exerting minimal impact on the level of appropriate total proteins. Additionally, AUM302 significantly inhibited the levels of c-Myc compared to DMSO- and TP-3654 treatments. Thus far, we have demonstrated that AUM302 more effectively

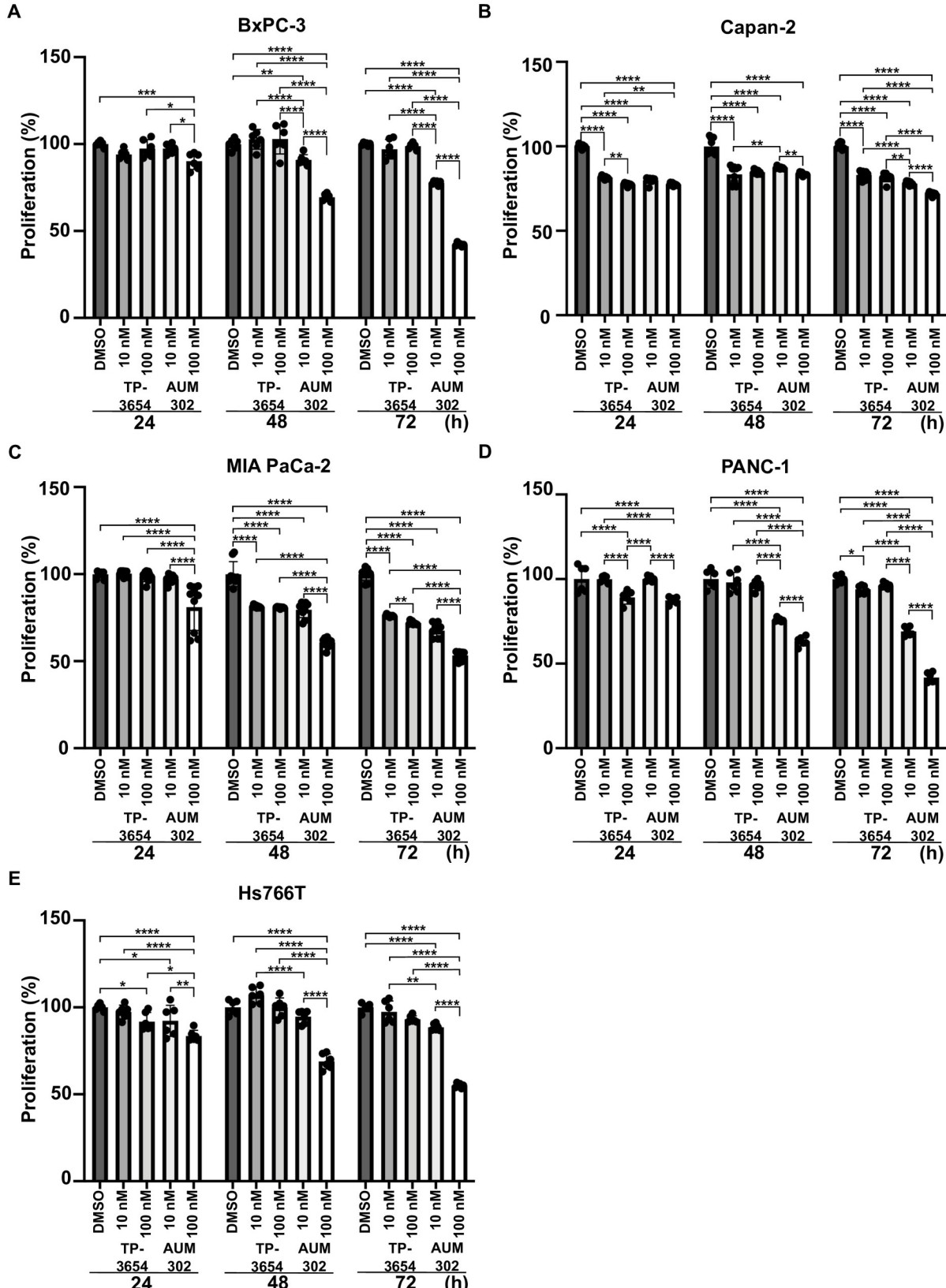

**Fig 2. AUM302 inhibits the proliferation of multiple pancreatic cancer cell lines.** The following pancreatic cancer cell lines, BxPC-3 (A), Capan-2 (B), MIA PaCa-2 (C), PANC-1 (D), and Hs766T (E), were treated with DMSO or TP-3654 (10 nM and 100 nM), or AUM302 (10 nM

and 100 nM). Cell count was determined 24, 48, and 72 hours after treatment using a cell counter. The measurement of the control (cells with DMSO) was defined as 100%. Data represent mean ±SD ($N = 6$). *p<0.05; **p<0.01; ***p<0.001; ****p<0.0001 calculated with two-way ANOVA.

inhibits the proliferation of pancreatic cancer cells. Importantly, in contrast to TP-3654, AUM302 significantly modified the progression of the cell cycle and induced apoptosis. It has been shown that PI3K plays a vital role in the development of chemoresistance to gemcitabine in pancreatic cancer, and inhibition of PI3K activity can reverse this resistance [52, 80–83]. Hence, we decided to determine whether treatment with AUM302 inhibits the growth of gemcitabine-resistant pancreatic cancer cells.

## AUM302 inhibits growth of gemcitabine-resistant pancreatic cancer cells

To assess the ability of TP-3654 and AUM302 to inhibit the growth of gemcitabine resistant pancreatic cancer cells we employed MIA PaCa-2 Gemcitabine Resistant (MIA PaCa-2 GemR) cell line. MIA PaCa-2 GemR cell line was continually grown in media supplemented with 50 nM gemcitabine. First, we grew MIA PaCa-2 GemS and MIA PaCa-2 GemR cell lines in the presence of TP-3654 and AUM302 at 10 nM, 100 nM, and 1 µM. We assessed the cell viability using Cell Titer-Glo 72 hours post-treatment. As shown in Fig 8, TP-3654 at 10 nM and 100 nM did not affect the growth of MIA PaCa-2 GemS and MIA PaCa-2 GemR cells (Fig 8A and 8B). The significant decrease in MIA PaCa-2 GemR cells viability was shown when cells were treated with 1 µM TP-3654 (Fig 8C). In contrast, even low doses of AUM302 (10 nM and 100 nM) significantly inhibited viability of MIA PaCa-2 GemR (Fig 8D and 8E). The effect was even more pronounced when cells were treated with 1 µM AUM302 (Fig 8F).

The effect of TP-3654 and AUM302 was additionally tested on the proliferation of MIA PaCa-2 GemR cell line over three days (Fig 9A). MIA PaCa-2 GemR cells were treated with at DMSO (vehicle) and 10 nM, 100 nM, or 1 µM of TP-3654 and AUM302 and cells were collected and counted 24, 48, and 72 hours post-treatment. As shown in Fig 9A, TP-3654 treatment did not affect the growth of MIA PaCa-2 GemR cells. In contrast, AUM302 reduces the growth of MIA PaCa-2 GemR cells at 72 hours at all tested concentration. Notably, treatment with 1 µM AUM302 almost completely inhibited the growth of MIA PaCa-2 GemR after 24 hours of treatment (Fig 9B). Furthermore, protein analysis was completed on MIA PaCa-2 GemR cells treated with DMSO, or TP-3654 or AUM302 at 10 nM, 100 nM, or 1 µM concentration for 24, 48, and 72 hours before total protein extraction. Western blot analysis demonstrated that components of the PI3K/AKT/mTOR signaling pathway are significantly inhibited with AUM302 compared to DMSO- and TP-3654-treated cells (Fig 9B–9D and S1–S3 Figs). Importantly, the inhibitory effects of AUM302 (100 nM and 1 µM) on the phosphorylation levels of mTOR, AKT, and S6 were noted already at 24-hour timepoint. In addition, the levels of non-phosphorylated counterparts of these proteins were downregulated after the treatment with 1µM AUM302 compared to other treatments. While treatment with TP-3654 resulted with significantly lower inhibition of the phosphorylation of mTOR, AKT, and S6, and downregulation of the total protein levels. Furthermore, we assessed the levels of c-Myc, and we showed that AUM302 at 1 µM concentration was able to significantly reduce its levels at three tested time points.

## Discussion

Pancreatic ductal adenocarcinoma (PDAC) is a lethal cancer often resistant to the widely used chemotherapeutic agent gemcitabine. Currently, limited options exist for the treatment of

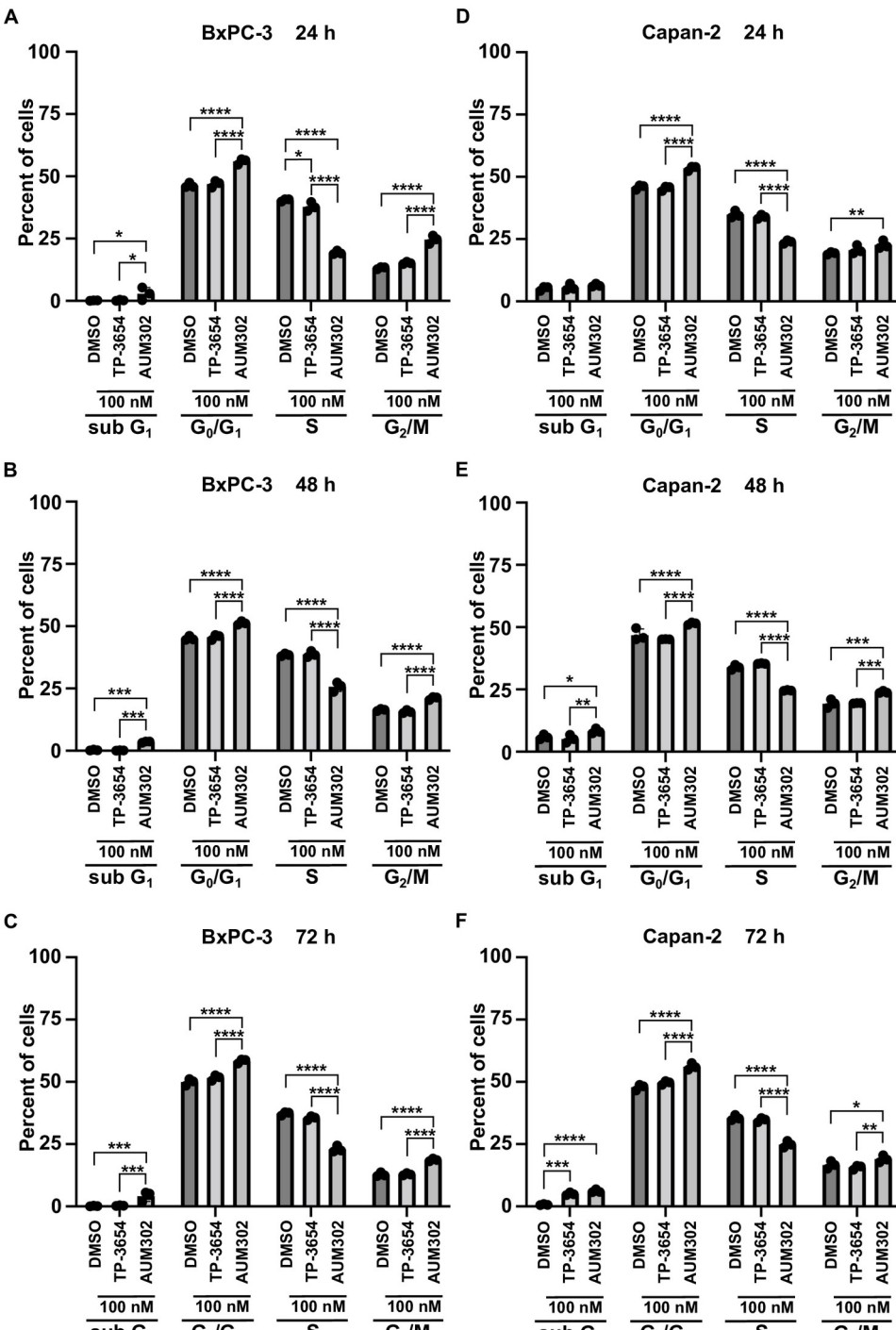

**Fig 3. AUM302 changes the cell cycle profile of BxPC-3 and Capan-2 pancreatic cancer cell lines.** Cells were treated with DMSO or TP-3654 (100 nM) or AUM302 (100 nM) for 24 (A and D), 48 (B and E), and 72 hours (C and F). Cells were stained with propidium iodide and analyzed by FACS analysis. Data are represented as mean ±SD, $N = 3$, *p<0.05; **p<0.01; ***p<0.001; ****p<0.0001 calculated with two-way ANOVA.

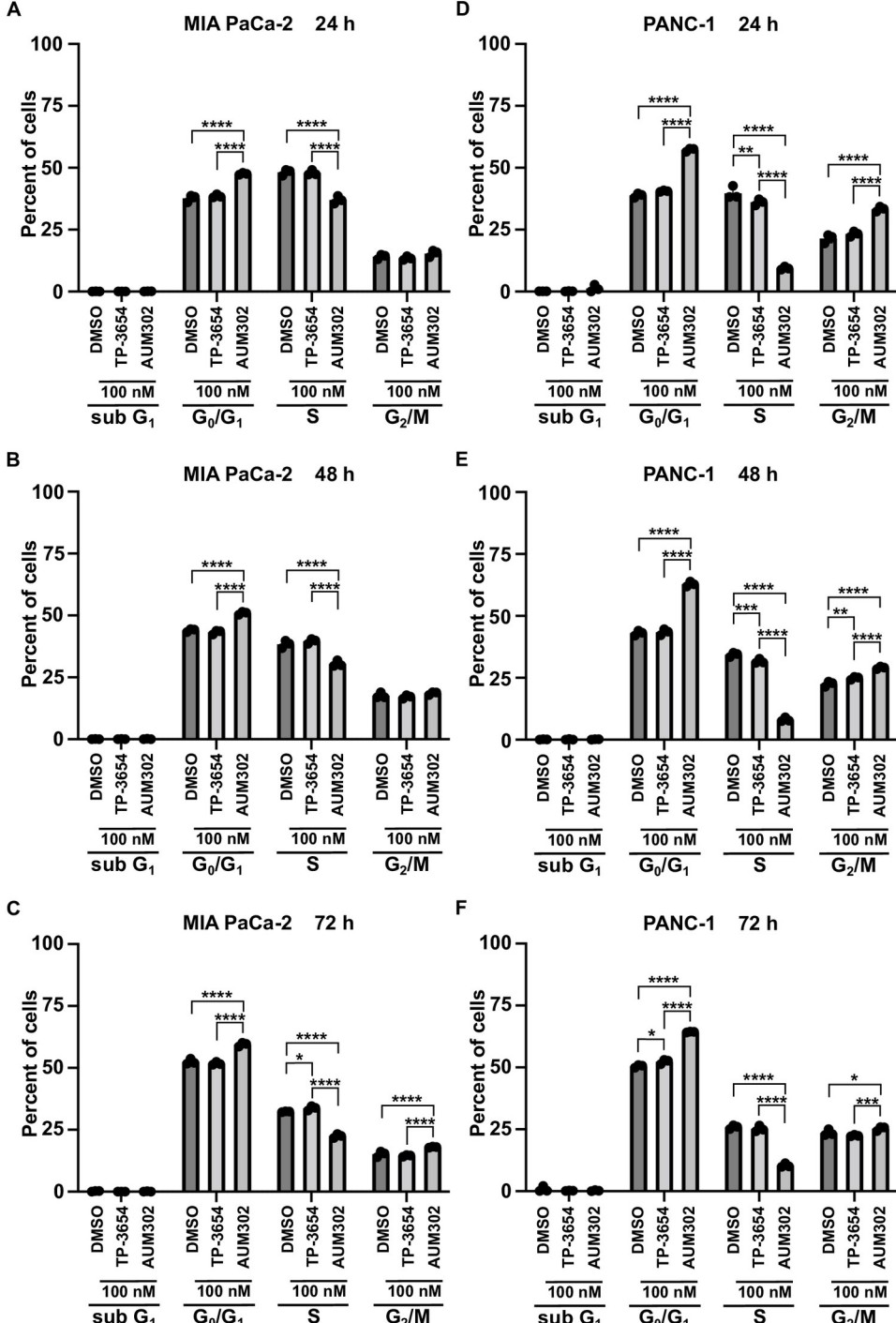

**Fig 4. AUM302 changes the cell cycle profile of MIA PaCa-2 and PANC-1 pancreatic cancer cell lines.** Cells were treated with DMSO or TP-3654 (100 nM) or AUM302 (100 nM) for 24 (A and D), 48 (B and E), and 72 hours (C and F). Cells were stained with propidium iodide and analyzed by FACS analysis. Data are represented as mean ±SD, $N = 3$, *$p<0.05$; **$p<0.01$; ***$p<0.001$; ****$p<0.0001$ calculated with two-way ANOVA.

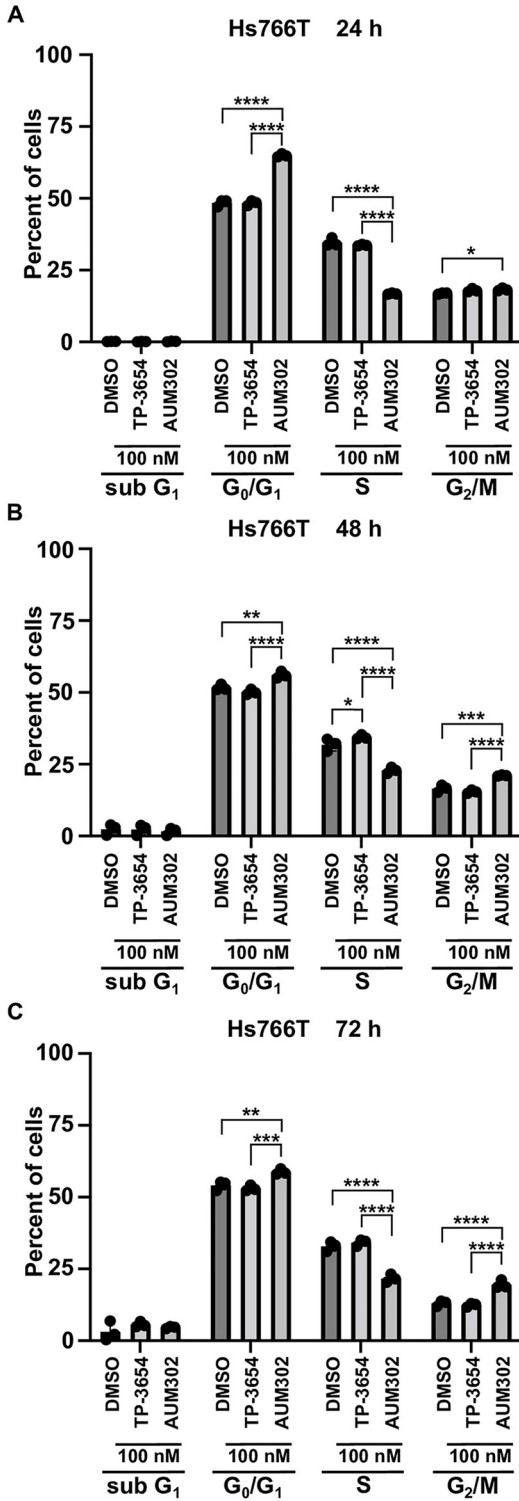

**Fig 5. AUM302 changes the cell cycle profile of Hs766T pancreatic cancer cell line.** Cells were treated with DMSO or TP-3654 (100 nM) or AUM302 (100 nM) for 24 (A), 48 (B), and 72 hours (C). Cells were stained with propidium iodide and analyzed by FACS analysis. Data are represented as mean ±SD, $N = 3$, *p<0.05; **p<0.01; ***p<0.001; ****p<0.0001 calculated with two-way ANOVA.

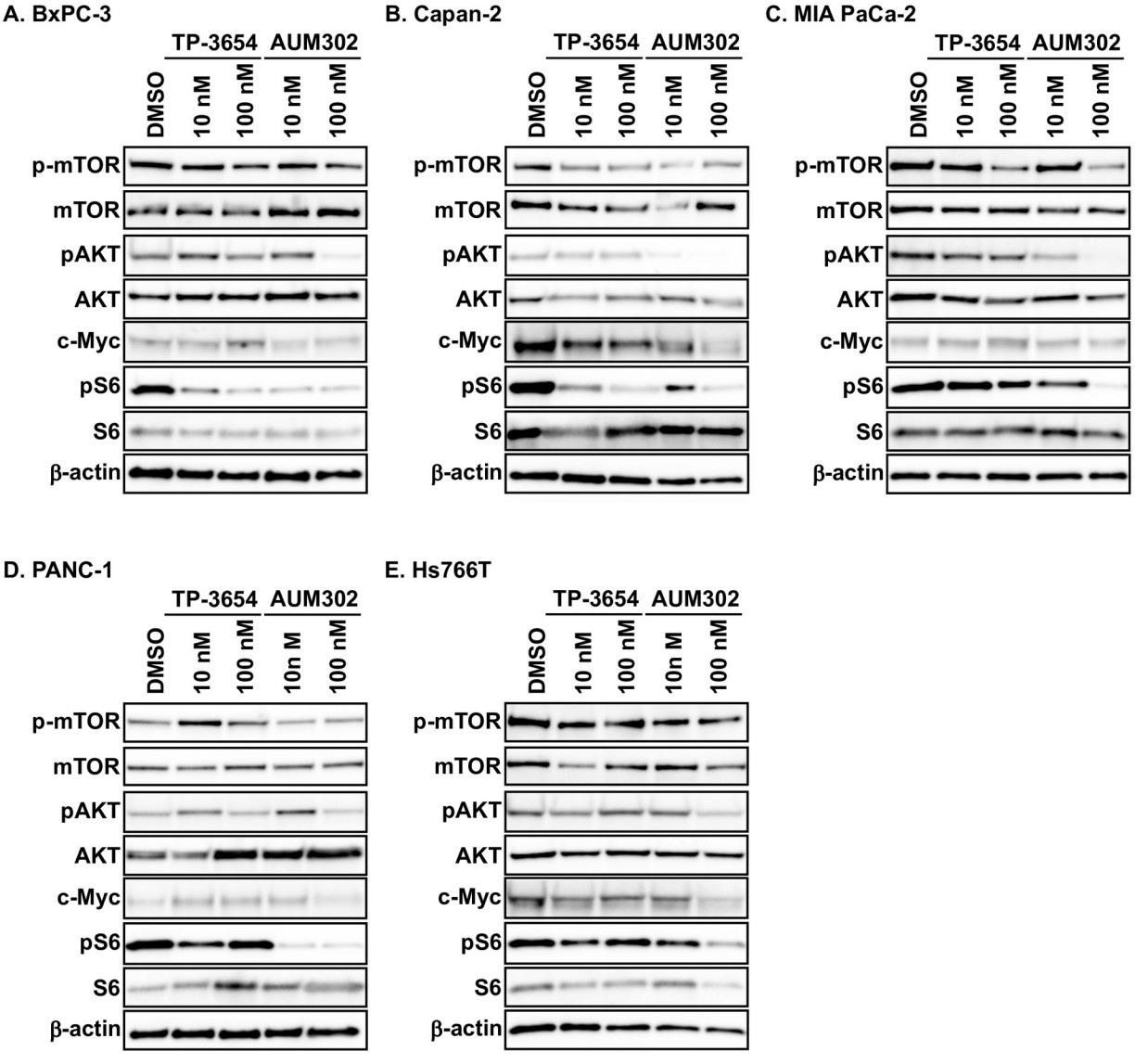

**Fig 6. AUM302 inhibits the cell signaling pathways regulated by PIM kinases and PI3K/mTOR pathway.** BxPC-3 (A), Capan-2 (B), MIA PaCa-2 (C), PANC-1 (D), and Hs766T (E) cells were treated with DMSO (vehicle) or TP-3654 (10 and 100 nM) or AUM302 (10 and 100 nM) for 24 hours.

PDAC. The surgery followed by adjuvant chemotherapy with FOLFIRINOX is available only to 10–15% of PDAC patients and provides an overall survival rate of four and a half years [84–86]. PDAC multidrug treatment provides between two to six months of benefit for patients with locally advanced or metastasis compared to single compound treatment [84, 85]. Consequently, studying PDAC molecular pathways gives rise to targeted therapies that may be a promising approach to treating pancreatic cancer [87]. Targeting PIM kinases has become a novel cancer therapeutic approach [88]. PIM inhibitors such as AZD1208, PIM447, and TP-3654 [16] are used against different cancers and have all entered the clinical stage [89]. A phase I dose-escalation study on 35 patients with solid tumors found that AZD1208 induced no functional response, although the PIM kinases were inhibited [90]. TP-3654 has improved

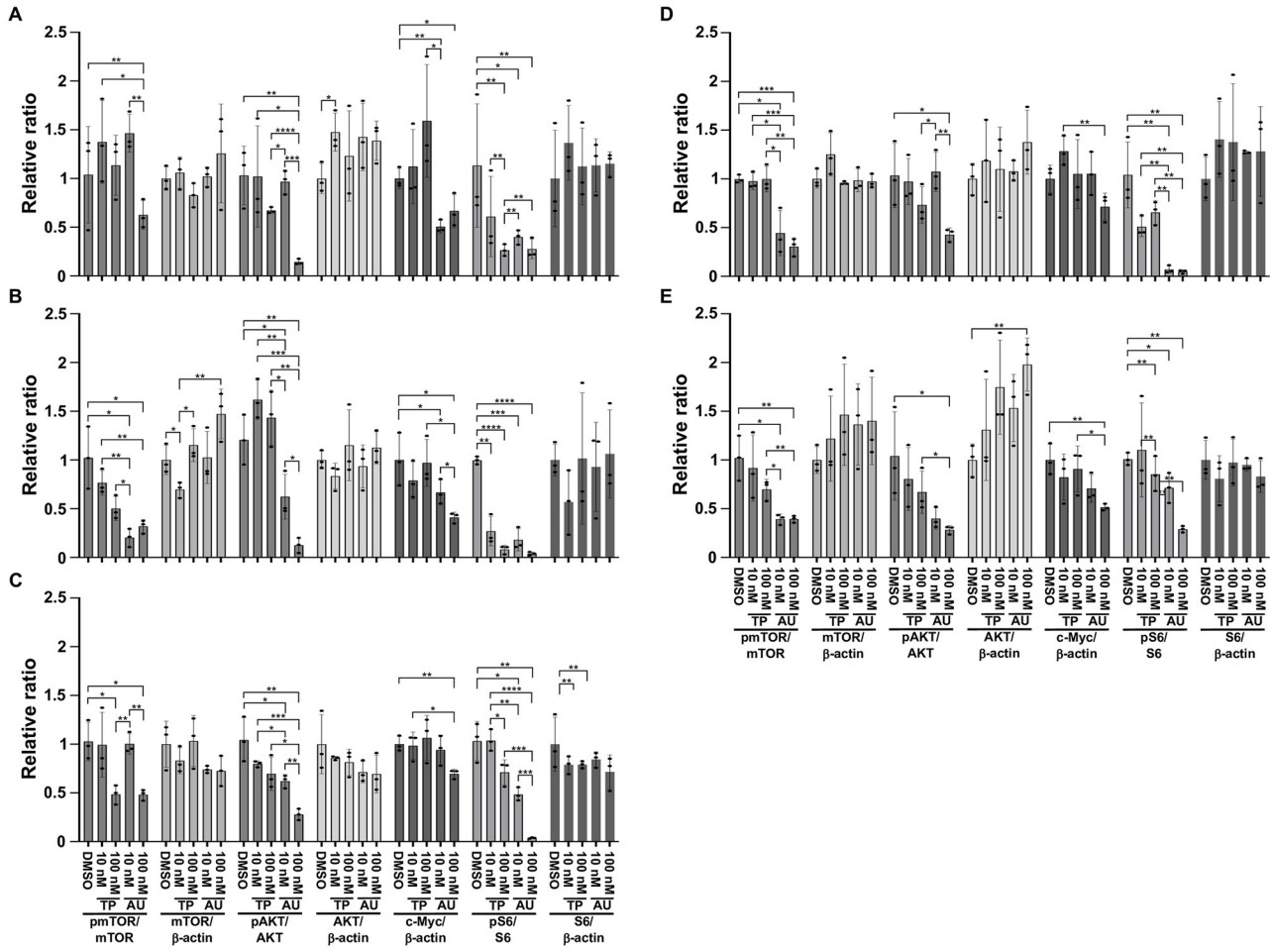

**Fig 7. Densitometry analysis of western blots results of protein regulated by PIM and PI3K/mTOR pathway.** BxPC-3 (A), Capan-2 (B), MIA PaCa-2 (C), PANC-1 (D), and Hs766T (E) cells were treated with DMSO (vehicle) or TP-3654 (10 and 100 nM) or AUM302 (10 and 100 nM) for 24 hours. Each experiment was performed in triplicate and the results are shown as mean ±SD ($N = 3$). Densitometry analysis was performed using FIJI software [61]. Statistical analysis was performed using the Student's test followed by an analysis of the normal distribution (Tukey's test). *p<0.05; **p<0.01; ***p<0.001; ****p<0.0001.

potency, and clinical findings suggest it can treat patients with heavily pretreated, relapsed, and resistant solid tumors. There is a strong rationale to investigate PIM inhibitors in combination with PI3K/AKT/mTOR pathway inhibitors, as their interactions drive cancer cell proliferation and cell survival [31].

PI3K inhibitors are a targeted therapy with limited success in treating PDAC but promising results when combined during pre-clinical studies [16]. Initially, pan-PI3K or PI3K/mTOR inhibitors such as GDC-0941 or BEZ235 were developed and tested in preclinical models. Studies showed that simultaneous inhibition of PI3K with GDC-0941 and MEK or PORCN inhibitors provides a synergistic effect [91–93]. Other studies showed BEZ235 enhanced response to the chemotherapy and antiangiogenic or pan-histone deacetylase inhibitors in PDAC treatment [94, 95]. However, resistance to PI3K inhibitors is possible, and PIM overexpression was found to be related to such resistance [96]. Furthermore, PIM mimics the effects of AKT, leading to cell cycle progression, cell survival, and growth [97]. mTOR is a link between the PIM and PI3K pathways, also responsible for cell survival. These pathways are so intertwined in cancer that a multikinase inhibitor may be an efficient approach.

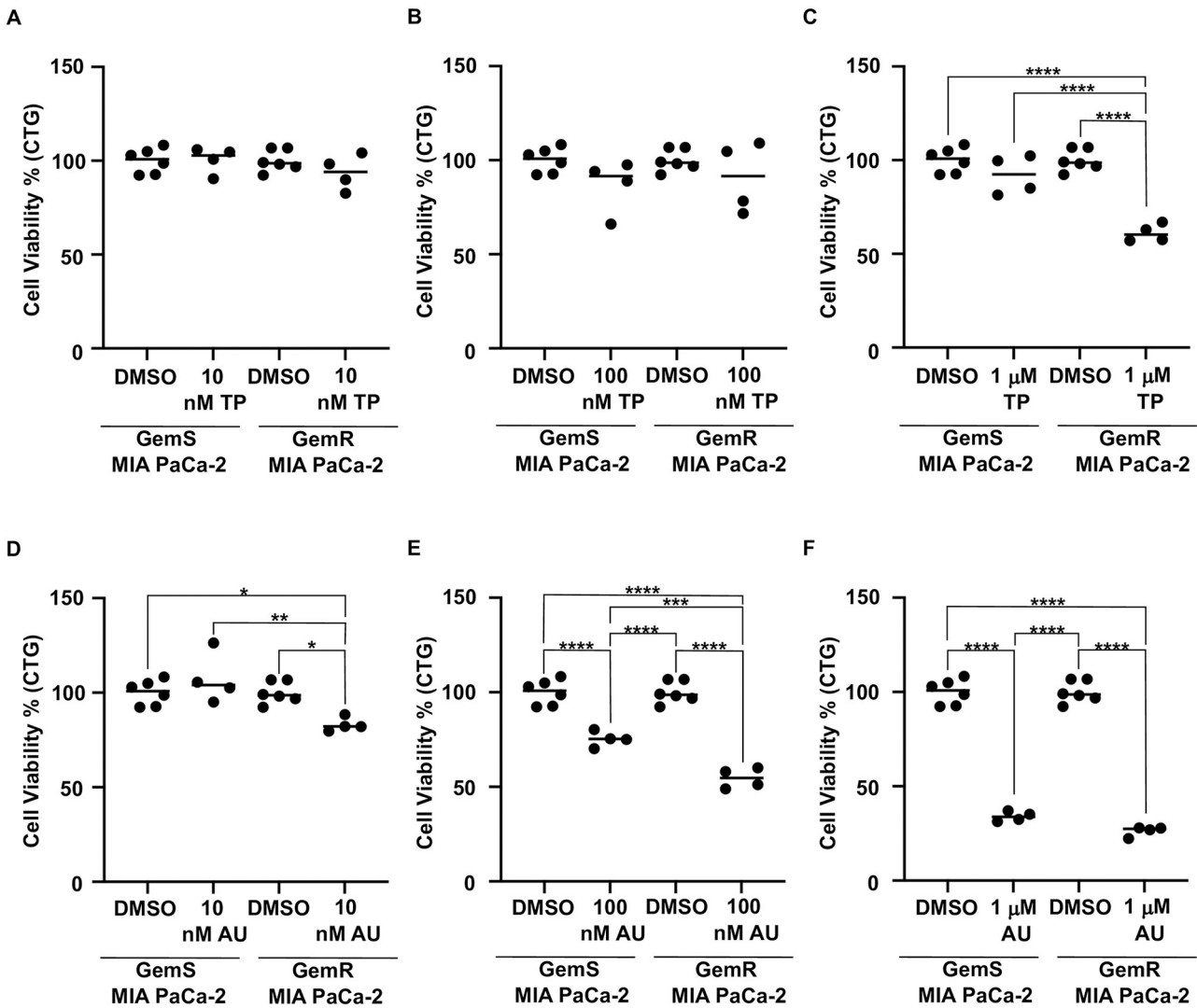

**Fig 8. AUM302 and TP-3654 decrease the cell viability of MIA PaCa-2 gemcitabine-resistant (GemR) cell line.** MIA PaCa-2 GemR cell line was treated with 10 nM, 100 nM, or 1 μM of TP-3654 (A, B, & C) or AUM302 (D, E, & F) twenty-four hours after seeding. Cells were treated with test compounds for 72 hours and cell viability was measured using Cell Titer-Glo. Data represents mean ±SD (N = 6 and N = 4). *p<0.05; **p<0.01; ***p<0.001; ****p<0.0001 calculated with two-way ANOVA.

Here, we investigated using a triple kinase PIM/PI3K/mTOR inhibitor, AUM302, compared to TP-3654 and gemcitabine in PDAC cell lines. Initially, we compared the efficacy of two known dual PI3K/mTOR inhibitors, GDC-0941 and BEZ235, with triple PIM/PI3K/mTOR inhibitors, TP-3654 and AUM302, to reduce the viability of PDAC cell lines. Our results showed AUM302 had lower $IC_{50}$ values in four tested cell lines (BxPC-3, Capan-2, PANC-1, and Hs766T) than other treatments. MIA PaCa-2 cell line was more susceptible to tested dual inhibitors than triple ones. It has been shown that the downregulation of *PIM1* by shRNA in MIA PaCa-2 cell lines decreases proliferation [98]. However, the treatment in the context of PI3K/mTOR inhibitors may not result in more efficient viability inhibition (Fig 1, Table 1). AUM302 potently inhibited growth in PDAC cell lines, generating $IC_{50}$ values in the low nanomolar range and significantly decreased cell viability. TP-3654 had $IC_{50}$ values in the

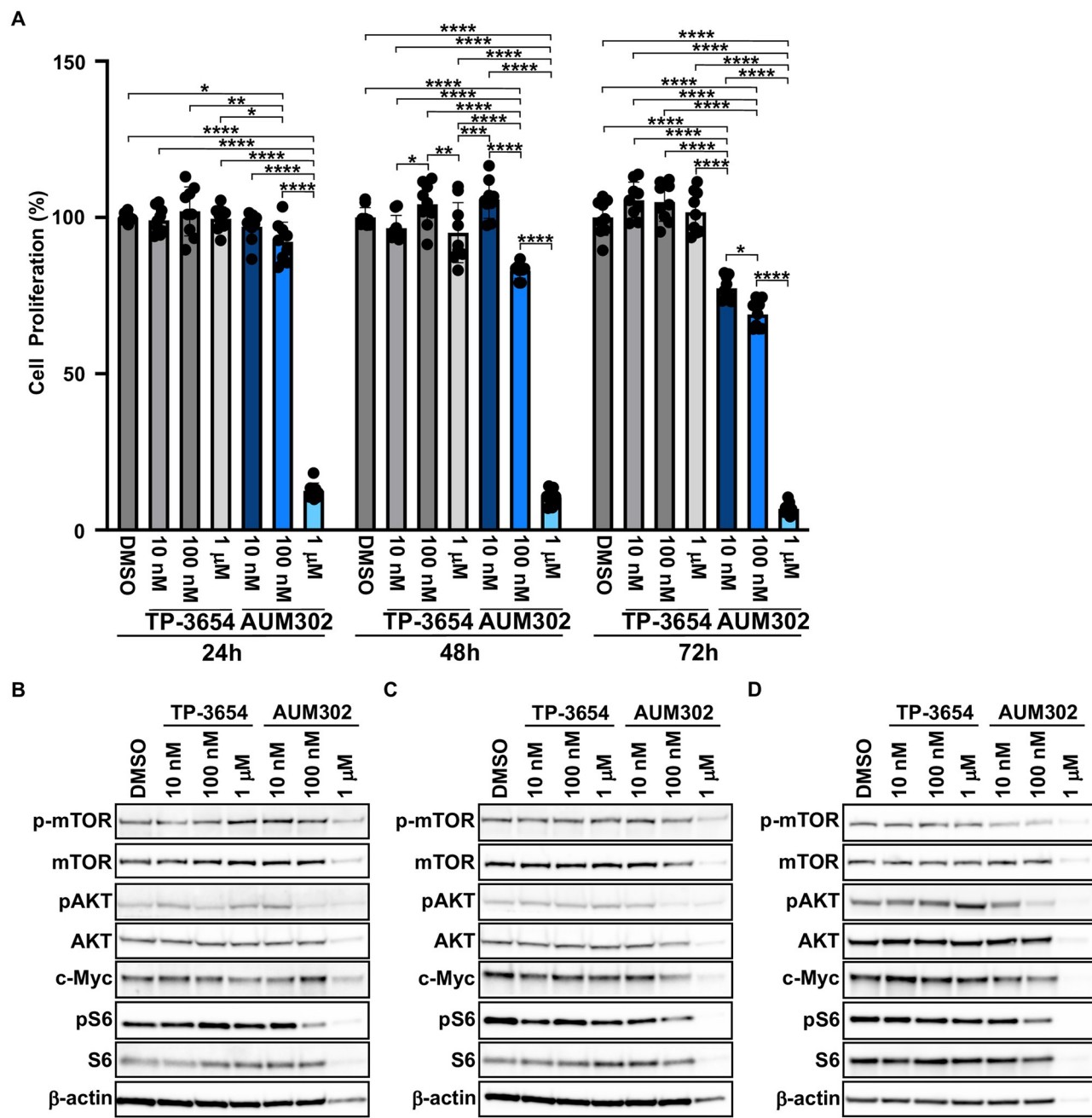

**Fig 9. AUM302 inhibits the proliferation of MIA PaCa-2 gemcitabine-resistant (GemR) cells and activity of multiple signaling pathways.** (A) MIA PaCa-2 GemR cells were treated with 10 nM, 100 nM, and 1 μM of TP-3654 or AUM302. Cell count was determined 24, 48, and 72 hours after treatment using a cell counter. The measurement of the control (cells with DMSO) was defined as 100%. Data represent mean ±SD ($N = 9$). *p<0.05; **p<0.01; ***p<0.001; ****p<0.0001 calculated with two-way ANOVA. (B—D) MIA PaCa-2 GemR cells were treated with DMSO (vehicle) or TP-3654 or AUM302 (10 nM, 100 nM, and 1 μM) for 24 (B), 48 (C), and 72 (D) hours and analyzed with western blot.

micromolar range and was less effective in altering the viability of PDAC cells, and gemcitabine even less so. Follow-up experiments comparing the impact of TP-3654 and AUM302 on cell proliferation and cell cycle showed that AUM302 is a more potent inhibitor of proliferation and blocker of the cell cycle progression than TP-3654 (Figs 2–5). Furthermore, AUM302

consistently decreased the number of cells in the S-phase and increased in $G_2$/M phase compared to TP-3654 treated cells and control. This may be due to inhibition of PIM1 activity, which regulates normal cell cycle progression, particularly at the $G_1$/S checkpoint [99]. The effect of AUM302 may be due to reduced expression of transcription factors like c-Myc, regulating cellular metabolism and protein translation through mTOR and AKT, regulating apoptosis through BAD, and decreasing the phosphorylation and activity of the ribosomal protein S6 [100]. Our results demonstrated that 24-hour treatment with AUM302 inhibited the phosphorylation of mTOR, AKT, and S6, while the total levels of the appropriate proteins were almost unchanged (Figs 6 and 7). In contrast, TP-3654 did not have a significant effect or demonstrated minimal impact on the expression levels of these proteins. In addition, we showed that the levels of c-Myc, an effector of the PI3K/mTOR pathway, were significantly decreased upon AUM302 compared to other treatments. This observation agrees with the previous studies demonstrating the downregulation of c-Myc levels upon inhibition of PI3K/mTOR in several cancer types [101–103]. Importantly, PIM1 was shown to phosphorylate c-Myc and increase its stability [40]. Thus, the downregulation of c-Myc in our model could be due to inhibition of kinase activity of PIM and an increase in c-Myc degradation.

We further investigated the effect of AUM302 and TP-3654 on gemcitabine-resistant cells using the MIA PaCa-2 gemcitabine-resistant cell line. Treatment with TP-3654 had no significant impact on the cell viability of gemcitabine-resistant cells until a concentration of 1μM, AUM302 did have a considerable effect at 10 nM, 100 nM, and 1 μM (Figs 8 and 9). Similarly, AUM302 significantly decreased cell proliferation of gemcitabine-resistant cells compared to vehicle- and TP-3654-treated cells. This may be due to AUM302 reducing the phosphorylation and thus the activity of mTOR, AKT, and S6 and decreasing the levels of c-Myc over three-day treatment (Fig 1 and S1–S3 Figs).

This is the first *in vitro* study demonstrating that AUM302 is an effective inhibitor of the PIM/PI3K/mTOR pathways and decreases PDAC cell viability. Its efficacious multikinase properties make it an advantageous approach to cancer therapy compared to kinase inhibitors like TP-3654 or dual PI3K/mTOR inhibitors. Notably, the compound can overcome gemcitabine resistance in PDAC cells. AUM302 is a potent inhibitor of PDAC cell growth *in vitro*, and our results suggest a clinical benefit in future research. However, PDAC is characterized by great genetic intra- and inter-heterogeneity [104]. In addition, treatment outcome also depends on the interaction between tumor cells and the microenvironment. In the case of PDAC, extensive fibrosis with little vascularization limits the drug's efficacy. Thus, to fully assess the effectiveness of AUM302, studies using models mirroring *in vivo* characteristics of PDAC, such as chemically-induced animal models and genetically engineered mice, patient-derived organoids, and xenografts, should be performed [105, 106].

## Supporting information

**S1 Fig. Densitometry analysis of western blots results of proteins regulated by PIM and PI3K/mTOR pathway in MIA PaCa-2 GemR cells treated with 10 nM, 100 nM, and 1 μM of TP-3654 or AUM302 for 24 hours.** Each experiment was performed in triplicate and the results are shown as mean ±SD ($N = 3$). Densitometry analysis was performed using FIJI software [61]. Statistical analysis was performed using the Student's test followed by an analysis of the normal distribution (Tukey's test). *p<0.05; **p<0.01; ***p<0.001.
(PDF)

**S2 Fig. Densitometry analysis of western blots results of proteins regulated by PIM and PI3K/mTOR pathway in MIA PaCa-2 GemR cells treated with 10 nM, 100 nM, and 1 μM of TP-3654 or AUM302 for 48 hours.** Each experiment was performed in triplicate and the

results are shown as mean ±SD ($N = 3$). Densitometry analysis was performed using FIJI software [61]. Statistical analysis was performed using the Student's test followed by an analysis of the normal distribution (Tukey's test). *p<0.05; **p<0.01; ***p<0.001.
(PDF)

**S3 Fig. Densitometry analysis of western blots results of proteins regulated by PIM and PI3K/mTOR pathway in MIA PaCa-2 GemR cells treated with 10 nM, 100 nM, and 1 μM of TP-3654 or AUM302 for 72 hours.** Each experiment was performed in triplicate and the results are shown as mean ±SD ($N = 3$). Densitometry analysis was performed using FIJI software [61]. Statistical analysis was performed using the Student's test followed by an analysis of the normal distribution (Tukey's test). *p<0.05; **p<0.01; ***p<0.001.
(PDF)

**S1 Raw images.**
(PDF)

## Acknowledgments

We would like to thank Research Flow Cytometry Core in the Department of Pathology, Stony Brook University for assistance with data analysis.

## Author Contributions

**Conceptualization:** Lee M. Graves, Antonio T. Baines, Agnieszka B. Bialkowska.

**Data curation:** Joseph F. LaComb, Agnieszka B. Bialkowska.

**Formal analysis:** Komala Ingle, Joseph F. LaComb, Agnieszka B. Bialkowska.

**Funding acquisition:** Agnieszka B. Bialkowska.

**Investigation:** Komala Ingle, Joseph F. LaComb, Agnieszka B. Bialkowska.

**Resources:** Antonio T. Baines.

**Supervision:** Agnieszka B. Bialkowska.

**Validation:** Komala Ingle, Agnieszka B. Bialkowska.

**Visualization:** Komala Ingle.

**Writing – original draft:** Komala Ingle, Joseph F. LaComb, Antonio T. Baines, Agnieszka B. Bialkowska.

**Writing – review & editing:** Komala Ingle, Joseph F. LaComb, Lee M. Graves, Antonio T. Baines, Agnieszka B. Bialkowska.

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
