## [Decision Letter · Decision Letter 0]

18 Jun 2023

PONE-D-23-13615AUM302, a novel triple kinase PIM/PI3K/mTOR inhibitor, is a potent pancreatic cancer growth inhibitorPLOS ONE

Dear Dr. Bialkowska,

Thank you for submitting your manuscript to PLOS ONE. After careful consideration, we feel that it has merit but does not fully meet PLOS ONE’s publication criteria as it currently stands. Therefore, we invite you to submit a revised version of the manuscript that addresses the points raised during the review process.

We look forward to receiving your revised manuscript.

Kind regards,

Wagdy Mohamed Eldehna, Ph.D

Academic Editor

PLOS ONE

“ABB DK124342 National Institutes of Health

https://www.nih.gov/

NO”

“We would like to thank Research Flow Cytometry Core in the Department of Pathology, Stony Brook University for assistance with data analysis. This work was supported by grant from the National Institutes of Health awarded to A.B.B. (DK124342).”

“ABB DK124342 National Institutes of Health

https://www.nih.gov/

NO”

Reviewers' comments:

Reviewer's Responses to Questions

**Comments to the Author**

1. Is the manuscript technically sound, and do the data support the conclusions?

Reviewer #1: Partly

Reviewer #2: Yes

Reviewer #3: Yes

2. Has the statistical analysis been performed appropriately and rigorously? 

Reviewer #1: Yes

Reviewer #2: Yes

Reviewer #3: Yes

3. Have the authors made all data underlying the findings in their manuscript fully available?

Reviewer #1: Yes

Reviewer #2: Yes

Reviewer #3: Yes

4. Is the manuscript presented in an intelligible fashion and written in standard English?

Reviewer #1: Yes

Reviewer #2: Yes

Reviewer #3: Yes

5. Review Comments to the Author

Reviewer #1: Bialkowska et al compare the effect in viablity, proliferation, cell cycle and biomarker modulation of a PIM-i with a triple inhibitor PIm/PI3K/mTOR in five pancreatic cell lines. Moreover, they compare their activity in one cell line resistant to gemcitabine. When they study the effect on biomarkers they treat the cells for 72h, a time point in which AUM-302 usually has a strong effect in viability or proliferation, so this data are not valid.

Major revision:

The authors should also include a PI3K/mTOR inhibitor to compare with and really demonstrate that the triple inhibitor is more potent

Minor revisions

1. Correct table 1. When a cell line is resistant to a drug de GI50 should be higher than the highest concentration tested

2. In order to be able to take any conclusions regarding the effect in biomarker modulation, on one hand the authors should provide a quantification of the bands and in the case of phosphorylation take into account the total levels of each corresponding protein.

Moreover, 72h is a very long time treatment to see effect in phosphorylation as compensatory mechanisms could take place.

3. In the case of resistant cell line, the highest concentration tested of AUM-302 for the evaluation of biomarkers has a very strong effect in proliferation at 72h then this results are not reliable. Shorter treatments should be included.

Reviewer #2: Running manuscript discuss a potential therapy of pancreatic cancer, UM302, as a novel triple kinase PIM/PI3K/mTOR inhibitor, is a potent pancreatic cancer. Although the point is very interesting, authors have to made major revision of their manuscript as recommended in the following points:

Methodology:

Why authors did not determine PIM1/2 inhibition to ensure the correlation with the demonstrated results and the correlation of inhibited molecular components of PIM/PI3K/m TOR

It is advisable for author to make a semiquantitative analysis of western plot bands to facilitate the interpretation of inhibited proteins

Results part:

Table one should include standard deviation of demonstrated IC50s, its title should be also more elucidated to be self-explanatory.

Quality of Figures 2-8 should be improved in their quality

Discussion:

Authors should reconstruct the discussion part to be more focused on discussing the demonstrated results and the correlation of inhibited molecular components of PIM/PI3K/m TOR.

Reviewer #3: Dr. Bialkowska and co-authors report that “AUM302, a novel triple kinase PIM/PI3K/mTOR inhibitor, is a potent pancreatic cancer growth inhibitor.” The authors should add the words in vitro to this title.

Overall, this is a well-executed and well-written study demonstrating that AUM302 inhibits the proliferation and viability of different human pancreatic cancer cell lines in culture and is capable of overcoming gemcitabine resistance in a gemcitabine-resistant MIA PaCa2 cell line.

Since the authors did not carry out in vivo studies with either intra-pancreatic or autochthonous models of PDAC, the authors should add a statement or two to the discussion section to indicate that such studies are necessary to determine whether AUM302 can be effective in vivo, and whether AUM302 is capable of altering in a beneficial manner the immune-excluding tumor microenvironment in this immune-cold cancer.

6. PLOS authors have the option to publish the peer review history of their article (what does this mean?). If published, this will include your full peer review and any attached files.

Reviewer #1: **Yes: **Carmen Blanco-Aparicio

Reviewer #2: No

Reviewer #3: No

---

## [Author Response · Author response to Decision Letter 0]

19 Sep 2023

September 12th, 2023

Editor and Reviewers PLOS ONE

Re: PONE-D-23-13615

AUM302, a novel triple kinase PIM/PI3K/mTOR inhibitor, is a potent pancreatic cancer growth inhibitor

Dear Editor and Reviewers,

We want to thank the Reviewers for their invaluable comments. We also acknowledge the concerns raised by the Reviewers and now provide a revised copy of the manuscript, considering all of the Reviewers' concerns. This revised manuscript addressed all the comments and suggestions the Editor and Reviewers provided. Please see our responses to your comments below.

Reviewer 1.

Comment 1. The authors should also include a PI3K/mTOR inhibitor to compare with and really demonstrate that the triple inhibitor is more potent.

Response 1. We agree with the Reviewer comment. We performed cell viability studies in BxPC-3, Capan-2, MIA PaCa-2, PANC-1, and Hs766T using two compounds, GDC-0941 and BEZ235, both PI3K and mTOR inhibitors. We included the results in Figure 1 and Table 1 and added an appropriate statement in the Results section.

Comment 2. Correct table 1. When a cell line is resistant to a drug de GI50 should be higher than the highest concentration tested.

Response 2. We provided IC50 values in nanomolar concentrations. We performed all experiments using low concentrations of AUM-302 of 10nM and 100nM as we have observed its significant effects using these concentrations on cell proliferation and cell cycle of tested PDAC cell lines.

Comment 3. In order to be able to take any conclusions regarding the effect in biomarker modulation, on one hand the authors should provide a quantification of the bands and in the case of phosphorylation take into account the total levels of each corresponding protein. Moreover, 72h is a very long time treatment to see effect in phosphorylation as compensatory mechanisms could take place.

Response 3. We agree with the Reviewer's comment. In this revised manuscript, we included the results of the 24-hour treatment of five tested PDAC cell lines with 10nM and 100nM concentrations of TP-3654 and AUM-302 in Figure 6 and the quantification of the western blot results in Figure 7.

Comment 4. In the case of resistant cell line, the highest concentration tested of AUM-302 for the evaluation of biomarkers has a very strong effect in proliferation at 72h then this results are not reliable. Shorter treatments should be included.

Response 4. We provided western blot results from 24-, 48- and 72-hour treatment in Figure 9 and appropriate quantifications of the western blot results in Supplementary Figures 1 through 3.

Reviewer 2.

Comment 1. Why authors did not determine PIM1/2 inhibition to ensure the correlation with the demonstrated results and the correlation of inhibited molecular components of PIM/PI3K/m TOR?

Response 1. The AUM-302 compound does not impact the levels of PIM1/2 on RNA or protein levels. The compound has been designed to affect their activity. Thus, the levels of PIM1/2 are unchanged, but their activities are decreased upon AUM-302 treatment.

Comment 2. It is advisable for author to make a semiquantitative analysis of western plot bands to facilitate the interpretation of inhibited proteins.

Response 2. We agree with the Reviewer's suggestions. We performed appropriate quantifications and included them in Figure 7 and Supplementary Figures 1 through 3.

Comment 3. Table one should include standard deviation of demonstrated IC50s, its title should be also more elucidated to be self-explanatory.

Response 3. Thank you for your suggestion. We modified the title of Table 1 and provided IC50 values in nanomolar concentrations with standard deviations.

Comment 4. Quality of Figures 2-8 should be improved in their quality.

Response 4. All figures were prepared in Adobe Photoshop according to the PLOS ONE requirements with at least 300-600 dpi. All figures are submitted as .tif files. In addition, we utilized Preflight Analysis and Conversion Engine (PACE) digital diagnostic tool recommended by PLOS ONE that tests the quality of prepared images and their compliance with the journal requirements. It could be that the journal provides low-quality figures for the primary review. We hope that the quality of the revised figures is adequate.

Comment 5. Authors should reconstruct the discussion part to be more focused on discussing the demonstrated results and the correlation of inhibited molecular components of PIM/PI3K/mTOR.

Response 5. We want to thank the Reviewer for this suggestion. We revised the Discussion section and added more information relevant to inhibiting PIM/PI3K/mTOR pathway components.

Reviewer 3.

Comment 1. The authors should add the words in vitro to this title.

Response 1. We modified the title of the manuscript to reflect its in vitro studies.

Comment 2. Since the authors did not carry out in vivo studies with either intra-pancreatic or autochthonous models of PDAC, the authors should add a statement or two to the discussion section to indicate that such studies are necessary to determine whether AUM302 can be effective in vivo, and whether AUM302 is capable of altering in a beneficial manner the immune-excluding tumor microenvironment in this immune-cold cancer.

Response 2. Thank you for this suggestion. We modified the Discussion section and added a statement emphasizing the necessity for in vivo studies to assess AUM-302 full impact on the PDAC microenvironment.

We confirm that neither the manuscript nor any parts of its content are currently under consideration or published in another journal. We changed the revised manuscript to ascertain compliance with the PLOS ONE journal's requirements.

We hope the incorporated changes will satisfy the Reviewers and Editor and render the revised manuscript suitable for publication. Thank you for being so considerate.

All authors have approved the manuscript and agree with its submission to PLOS ONE.

Please feel free to contact me if I can be of further assistance,

Sincerely Yours, 

Agnieszka B. Bialkowska, PhD

Associate Professor

Renaissance School of Medicine at Stony Brook University

Department of Medicine

GI Translational Research Lab

HSC-T17 Room 090

Stony Brook, NY 11794-8176

Phone: (631) 638 2161

Email: Agnieszka.Bialkowska@stonybrookmedicine.edu

---

## [Decision Letter · Decision Letter 1]

25 Oct 2023

AUM302, a novel triple kinase PIM/PI3K/mTOR inhibitor, is a potent in vitro pancreatic cancer growth inhibitor

PONE-D-23-13615R1

Dear Dr. Bialkowska,

We’re pleased to inform you that your manuscript has been judged scientifically suitable for publication and will be formally accepted for publication once it meets all outstanding technical requirements.

Kind regards,

Wagdy Mohamed Eldehna, Ph.d

Academic Editor

PLOS ONE

Additional Editor Comments (optional):

Reviewers' comments:

Reviewer's Responses to Questions

**Comments to the Author**

1. If the authors have adequately addressed your comments raised in a previous round of review and you feel that this manuscript is now acceptable for publication, you may indicate that here to bypass the “Comments to the Author” section, enter your conflict of interest statement in the “Confidential to Editor” section, and submit your "Accept" recommendation.

Reviewer #1: All comments have been addressed

Reviewer #3: All comments have been addressed

2. Is the manuscript technically sound, and do the data support the conclusions?

Reviewer #1: Yes

Reviewer #3: Yes

3. Has the statistical analysis been performed appropriately and rigorously? 

Reviewer #1: Yes

Reviewer #3: Yes

4. Have the authors made all data underlying the findings in their manuscript fully available?

Reviewer #1: Yes

Reviewer #3: Yes

5. Is the manuscript presented in an intelligible fashion and written in standard English?

Reviewer #1: Yes

Reviewer #3: Yes

6. Review Comments to the Author

Reviewer #1: Authors have addressed all my comments properly. They just only need to rewrite lines 244 and 245 regarding the time treatment, as for DMSO they refer 72h and for treatment inhibitors 24h. all time points should be 24h.

Reviewer #3: Dr. Bialkowska and co-authors have addressed my concerns and the paper in my opinion presents important new information in a valid manner and is acceptable for publication.

7. PLOS authors have the option to publish the peer review history of their article (what does this mean?). If published, this will include your full peer review and any attached files.

Reviewer #1: **Yes: **Carmen Blanco-Aparicio

Reviewer #3: No

---

## [Editor Report · Acceptance letter]

31 Oct 2023

PONE-D-23-13615R1 

AUM302, a novel triple kinase PIM/PI3K/mTOR inhibitor, is a potent *in vitro* pancreatic cancer growth inhibitor 

Dear Dr. Bialkowska:

I'm pleased to inform you that your manuscript has been deemed suitable for publication in PLOS ONE. Congratulations! Your manuscript is now with our production department. 

Kind regards, 

on behalf of

Dr. Wagdy Mohamed Eldehna 

Academic Editor

PLOS ONE